# Decolonising an Irish Surname by Working the Hyphen of Gene-Ealogy

Esther Fitzpatrick [1,*] and Mike Fitzpatrick [2]

1   Faculty of Education, University of Auckland, Auckland 1023, New Zealand
2   Independent Researcher, Auckland 0604, New Zealand; mike@fitzpatrickclan.org
*   Correspondence: e.fitzpatrick@auckland.ac.nz; Tel.: +64-272201493

**Abstract:** The surname Fitzpatrick is readily identified as Irish. Until recently, the traditional Fitzpatrick surname narrative was of a medieval super-progenitor named Giolla Phádraig. His offspring, the eponymous Mac Giolla Phádraig, it was said, somehow came to dwell in every Irish province; yet this is an Irish surname myth that works to erase the history of ancient 'Fitzpatrick' clans. This article demonstrates how deconstructing the surname Fitzpatrick, through working the hyphen of gene-eaology, is a practice of decolonisation. Via genetic data and archival records, dominant clan identities are disrupted, while connections with lost clans are re/membered. Critical analysis dismantles the dominant narrative imposed by colonial strategies and reconnects people with kinship groups and forgotten forebears. Questions arise from the deconstruction of an Irish surname. How might new clan identities be imagined, and how is losing a dominant surname narrative negotiated?

**Keywords:** clan; critical; decolonisation; DNA; family history; Irish; surname

## 1. Introduction

Researching family history and its relationship to individual and clan identity is a tricky business. The problematics of disrupting narratives of individual identity linked to cultural associations remain, since individuals often define themselves through engaging deeply with identity stories (Ashcroft 2001; Hall 1996). These narrative identities are crafted over time, a 'rich tapestry' of stories that create connections with historical, communal, and memory-based threads (Farquhar 2010). In a previous article, we discussed how, through Y-DNA data findings and a critical review of archival data, there was a disruption of narratives pertaining to the surname Fitzpatrick (Fitzpatrick and Fitzpatrick 2020), which is borne by several different Irish clans. We asked if DNA could result in an upheaval of an individual's identity, what then would be the impact on the identity of entire clans? In this article, we further our utilisation of DNA and archival data to re/member lost ancient Fitzpatrick clans and consider the impacts of decolonising dominant clan narratives.

There has been a proliferation of published research critiquing family history research with the rise and popularity of genealogy (Nash 2008). As keen genealogists interested in the affordances of DNA research, we seek an understanding of how we navigate problematics yet simultaneously recognise the richness of material for creating connections and, importantly, developing a decolonising strategy. Through the process of mapping Y-DNA data with archival stories, lost ancient clan identities and clan connections to land and place are re/membered (Connerton 2009), countering the colonial practice of renaming peoples and places and erasing indigenous culture (Wolfe 2006).

We first position the complexity of contemporary Irish ethnic/cultural identity with decoloniality. We then use the notion of gene-eaology by working the hyphen as an ongoing struggle between two 'identities' (DNA and archival). Next, we briefly discuss notions of kinship and clan identity in the Irish context; problematics are identified by describing the limits of Y-DNA when investigating clan identity. A description of DNA research focusing

on the Irish surname Fitzpatrick is provided to set the context of the focus, followed by a discussion of the method of generating Y-DNA data in conversation with archival data. We then provide narratives of how this work has been actualised through groups of people with the same Fitzpatrick surname, who after robust Y-DNA and archival research, we learn have distinctly different clan histories. Lastly, we uncover some 'lost' Fitzpatrick clan townlands.

## 2. The Difficult Knowledge of Y-DNA

When engaging critically in Irish clan research, we are ultimately engaging in a conversation with 'difficult knowledge'. Britzman (2013) talks about difficult knowledge in relation to disrupting the ignorance of racial and ethnic stereotyping and bias. For descendants of diasporic Irish peoples, there is sometimes a tension between histories of being colonised and being complicit in the colonising project (Nash 2008). What a ghostly reminder, therefore, for those who claim Irishness, are the memories of ongoing waves of colonisation by Normans and English since 1169 AD and more recent emigration movements, which create what McVeigh and Rolston (2021) describe as a 'colonial present' with cultural associations of 'mixedness' (native and settler origins).

Issues with claiming Irish identity are complicated through centuries of strategic moves of colonisation and the impacts of an ongoing colonial project that inherently sought to eliminate and dispossess the indigenous Irish. The colonial settler instilled systems and structures to establish themselves and dominate the physical and social landscape (Wolfe 2006). Over time, the problems of authentic indigenous Irish were compounded through assimilation practices, such as intermarriage, loss of land, language, culture, and religion, and for many Irish the connection to a clan was lost.

Y-DNA thrusts new knowledge of clan identity into the conversation. Tricia Logan (2015, p. 435) contends that "when language, spirituality, ontology, educational systems and family structures are threatened or destroyed, the culture of people disintegrates along with the lives of people in the group or 'genos'". Further, Bell (2014) argues that as indigenous populations are absorbed into settler society, they are simultaneously culturally and genetically inauthentic. In this context, the significant advances in Y-DNA research mark a pivotal moment in time with respect to how ethnic/cultural identities are both understood and mobilised by individuals and groups. For Irish clans, it provides a way to potentially map out the complexity of clan cultures over time and place.

Using Y-DNA to identify Irish clans is nothing new. Compelling studies have analysed the Y-DNA of those with proven pedigrees, such as the late Sir Conor O'Brien, the eighteenth Baron Inchiquin, and linked them to archival records and numerous others of the clan (Wright 2009; Swift 2014). Less compelling are non-critical studies of those who attempt to make deep associations with ancient lineages using only modern surname occurrences in a location where a clan once dwelt. Often Y-DNA is used to declare, typically in a non-scholarly arena, a particular Y-DNA haplotype as representative of the descendants of a 'famous' clan and linked, via popular pedigrees, to an ancient Irish chieftain. Nash (2008) observes that genealogists have a reputation for being chasers of ancestors of note. Further, ideas of connection to an imagined 'orginal' or authentic Irish indigenous past persist (White 2016).

However, genealogy among the Irish demands far more than simply linking a Y-DNA haplotype with an ancient surname and claiming a noble pedigree. For one thing, Y-DNA surname studies have disrupted some dominant surname narratives and uncovered a complexity of surname adoption that is not easy to unravel. Hence, working the hyphen in gene-ealogy represents an ongoing struggle between DNA and archival identities. There is also recognition that for some, genealogy findings can 'cut deep' and cause much angst at both a personal and a clan level (Shapiro 2019; Fitzpatrick and Fitzpatrick 2020). Yet, amidst the upheaval, we argue that decolonising works are possible among Irish peoples via gene-ealogy.

### 3. Working the Hyphen

Applying the term gene-ealogy (Abel and Frieman 2023), we bring attention to how DNA data can be used in conversation with archival data. We creatively employ Fine's (1994) notion of 'working the hyphen' to engage critically with the different identity narratives and how "understanding that we are all multiple in those relations" and "how these 'relations between' get us 'better' data" (p. 72). Putting to work Y-DNA data and archival research to connect people to Irish clans, we argue, works the hyphen in gene-ealogy, i.e., through DNA research in conversation with archival research, where reconnections to clans, clan narratives, and places can be reimagined.

Identifying a hyphen—a place of connection between Y-DNA and archival data— is first. Y-DNA data is not enough to even begin to articulate clan identity. Significant is how Y-DNA might forge a connection to 'place', such as the townlands, parishes, and counties where people lived, yet the memory of which may be lost. Engaging in gene-ealogy enables us to re/member these histories, respecting the complexity of such identity stories, to analyse difficult knowledge and unsettle the dominant stories of homogeneity, and reveal the practice of assimilation. We argue here for the productive potential of coming to know how people are connected to these complicated pasts as a way to map the dynamic landscape of familial and kinship relations over time, cognisant that a connection to an ancient clan may turn out to be full of surprises, not all easy to accept.

For many people, the practice of gene-ealogy has transformed and challenged their understanding of family and identity. Abel and Frieman (2023) describe how 'gene-ealogies' "have proved to be a potent arena for the negotiation of identity, belonging, and authority— over both the past and the future" (p. 15). Likewise, Blom (2022) describes how DNA and archival data provide a "potential in-between space where bio-historic-cultural contents are negotiated with previous knowledge/experiences" (p. 334). The working of the hyphen in gene-ealogy research resonates with Ellsworth's (2005) "in-between or transitional space" and ways in which the "past and present inform each other" through an ongoing negotiation of self and others. Blom (2022) describes these becomings as "frayed edges of identity" that have "potential for social/cultural transformations" (p. 338). Importantly, DNA research demonstrates the connections between and across peoples, rather than definite identities, providing "material which can be incorporated into more complex narratives of identity" (Strand and Källén 2021, p. 524). Although

> arguably humans in general, relish creating categories and organization of people into groups . . . the narratives of sameness/difference presented by GAT [genetic ancestry testing] results, argue for an interconnected dynamic web that ebbs and flows (Blom 2022, p. 352).

Gene-ealogical work with a scholarly focus on the dynamic complexity of these ebbs and flows, over time and space, engages in disrupting dominant linear 'genealogical' narratives.

### 4. The Tricky Business of Irish Identity

Through a discussion on clan identity in Ireland, we consider the potentiality of creating connections as collectives rather than individualistic identity structures and how theories of kinship might inform notions of 'genetic' clanship as dynamic in productive ways. Irish clans are an expression of kinship. "Kin-making is a key part of how humans structure their relations with each other, their wider community, and the non-human world" (Frieman and Brück 2021, p. 47). Itao and Kaneko (2022) describe the evolution and dynamic process of kinship structures and how belonging to a "cultural association, or clan" is determined by "different 'rules' and relationships". People belong to these clans through shared "symbolic traits inherited through some descent lines" yet belong as cultural, not necessarily biological, kin (p. 2). Likewise, Frieman and Brück (2021) describe kin relations as constituted by "shared values and shared experience, as well as by shared cultural or biological lineage" (p. 47). For many societies, including Irish clan culture through fostering, adding others who are not 'blood' related as kin has a long history

(Costa et al. 2021). Kim TallBear (2013), indigenous American scholar, critiques how kinship is often

> profoundly influenced by the language of blood" reminding us to attend also to what is coded in our DNA or our genetic memory . . . [to] do so in a very particular social and historical context, one that entangles genetic information in a web of known family relations, reservation histories, and tribal and federal-government regulations. (p. 13).

Irish clan identity is political. For many settlers who have forgotten their origin narratives, an association with Irishness has the appeal of being distinctive (Nash 2008). Traditionally, ancient pedigrees were the primary source of locating connection to a clan. The still popular and oft cited John O'Hart, who wrote on the 'Origin and Stem of the Irish Nation' (1892), described the familial patriarchal structure of Ireland as a "civilization of clans" (p. xvi). Each clan had a chief, and pedigrees of 'noble' chiefs were charted and "established a hierarchy of worth" (Nash 2008, p. 129). Often families in a clan would share the surname of the chief, so for those searching for connection, the ancient pedigrees appear as a gold mine; however, identity and surname origin are not synonymous. Irish surname origins are not always connected to direct patrilineal genetic origins, and neither are genetic origins always connected to single surnames (Fitzpatrick and Fitzpatrick 2020). Nash (2008) further describes how the

> assumption that a specific family name defines a clan misrepresents the Gaelic system of septs [i.e., clan sub-groups], in which, . . . distinct septs, each with their own chief, can share a surname". In turn, "this troubles the assumption that a Gaelic surname in the present corresponds to descent from a specific clan in the past". (pp. 1743–44)

Clan identity is also explicitly tied to 'place', to historically occupied 'territories' (Basu 2005). O'Hart (1892) describes the importance to an Irish clan "of the soil of which was occupied" (p. x). Tuck and Yang (2012) describe how through colonisation, "settlers make Indigenous land their new home and source of capital" and disrupt "Indigenous relationships to land" (p. 5). After more than 850 years of colonisation in Ireland, the 'native', and 'settler' dichotomies remain unresolved, particularly in Northern Ireland (McVeigh and Rolston 2021). Identity through cultural association, such as a clan or tribe, is significant to how many people groups define themselves, and is particularly important for indigenous people seeking to decolonise and reclaim their identity narratives and reconnect to a place. These interwoven connections of place, narrative, and clan identity are referred to by Basu (2005) as fluid and eclectic "'changing responsive institutions' able to adapt to unfolding sociopolitical and economic conditions" (pp. 127–28).

We argue that colonisation processes worked to erase clan identity through renaming people groups, renaming places, and erasing connections to language, people, and place. Gene-ealogical work has the potential to attend to ruptured family narratives and reconnect kin with purposeful narratives (Nelson 2016). Our purpose aligns with Ford and Strauss's (2019) aim that

> in our conversations, we want to move beyond guilt, shame, and blame to imagine possibilities that could break open the many facets of our national story, acknowledge each other's respective truth(s), and heal. Perhaps we can even find a collective meaning in our shared legacy" (p. 227).

The limitations of Y-DNA research are understood. First, the problematics with how genealogical research re-reifies paternal histories through "geneticiz[ing] ethnicity and refigur[ing] both nation and diaspora as fundamentally communities of masculine, patrilineal and Gaelic descent" (Nash 2008). The ongoing practice of neglecting female ancestors to focus on 'founding fathers' and myths of heroic powerful men is further reinforced when attentive only to Y-DNA (Scully 2018; Nash 2015). Second, there can be a tendency to neglect other genetic histories and materials that inform identity. Third, connections that are not through birth, human and non-human, are generally not considered. Last, through

the practice of making connections, genealogical work also has the potential to disconnect, to "suggest not belonging as well as belonging within the diasporic clan" (Nash 2008, pp. 3515–16), where "continued risks can be seen in the homogenizing and exclusionary effects of genealogical incorporation" (Nash 2017, p. 2554).

Nevertheless, we recognise the value of Y-DNA data for sociology, anthropology, and other disciplines. As Barnes and Swain (2012) argue, there is an enormous range of genealogical data available that scholars can access. How, then, do we create a critical methodology that provides a decolonising approach to Irish clan history? This article draws on Y-DNA research as a tool to inform, not as an absolute truth of identity, recognising the complexity of identity formations, especially when probing into ancient clan histories. Instead, we attempt to derive new gene-ealogical narratives to disrupt dominant ancient pedigree narratives that have worked to erase particular stories that did not fit with those deemed 'worth/y'.

## 5. Dispelling the Myth of the Fitzpatrick Surname

The challenge of a dominant clan narrative that holds sway and the loss of clan narratives due to colonisation is encountered by those researching the surname Fitzpatrick (Fitzpatrick and Fitzpatrick 2020). Irish surname texts, such as those of O'Hart (1892), give the impression that all modern-day Fitzpatricks took their name from a single famous man, then having the surname Mac Giolla Phádraig of an Ossory (effectively modern-day Co. Kilkenny and Co. Laois) clan, whose ancestors were dispersed into every Irish province. This is a modern-day myth, and gene-ealogical research has provided a much different narrative for Fitzpatricks. For example, data from the 1901 Census of Ireland (The National Archives of Ireland 1901) shows the occurrence of the surname Fitzpatrick in every Irish county and numerous outside Ossory. Excluding Dublin, the five most popular for the surname were Cavan (12.5%), Laois (11.9%), Down (7.6%), Cork (6.3%), and Clare (5.0%). On this basis alone, a single ancestor theory is difficult to accept, and genetic genealogy has been able to peel back the skin of demographic data showing that Fitzpatrick is a surname that came to be taken by several distinct kindred lines.

Setting aside the problems associated with a clan definition based solely on paternal lineage is important since value can still be gained in that way, even with its limitations. Here, paternal lineages are identified by a Y-DNA SNP (single-nucleotide polymorphism), which arose ca. 850–1700 AD, i.e., after the emergence of surnames in Ireland (Ó Murchadha 1999), but no later than the cessation of the clan system in Irish society, defined by Clans of Ireland (2023), an authoritative body that registers Irish clans, as 1691 AD.

The Fitzpatrick DNA project, which is hosted at Family Tree DNA and has approximately 750 members (https://www.familytreedna.com/groups/fitzpatrick/about, accessed on 30 June 2023), has so far enabled the identification of at least 14 such lineages, or clans, that demonstrate a correlation between such a Y-DNA SNP and the surname. For several of the lineages, the correlation is strong. For others, a lesser degree of surname–SNP specificity indicates complexity since close, paternally related kindred groups came to bear unrelated surnames. While it is clear that 'Celticness' cannot be defined by genetic descriptors (ref), most Fitzpatrick lineages are 'genetically Celtic', according to a widely accepted definition among genetic genealogists that categorises the Celtic Y-DNA haplotype as R-L21 (Lucotte 2015).

Among these 'genetic Celts', some can be further defined by Irish dynasty, such as the Fitzpatrick Dál gCais, who are described in detail later. However, other 'Celtic' Fitzpatrick clans, such as those of haplotype R-BY12234, though numerous, have yet to be dynastically linked, and may never be. And 'non-Celtic' Fitzpatrick clans, which are very distant from R-L21, are also well represented on the DNA project. Additionally, regionally, Fitzpatrick surname representation may be by a single clan, but some regions have more; notably, there are at least four distinct Fitzpatrick clans of Bréifne, a medieval kingdom that encompassed most of Co. Cavan, much of Co. Leitrim, and portions of Co. Sligo, Co. Fermanagh, and Co. Meath.

In all this, the overriding consideration is that the genetic definition of Fitzpatrick clans should not be exclusive. In addition, for singletons, i.e., those who do not have a Y-DNA match with any other Fitzpatricks within the aforementioned timeframe, the absence of a Y-DNA match does not mean they do not have a Fitzpatrick clan origin. For example, it may simply be that other clan members have yet to be found, or that the connection came from a maternal line. For Fitzpatricks, the value of Y-DNA investigation comes because it can be used alongside critical archival research as a method for dispelling the myth of an origin from a single eponymous Mac Giolla Phádraig, transforming a surname group into multiple clans with individual narratives, and enabling a richer, and not diminished, connection to an associated identity with the surname Fitzpatrick. If marked Y-DNA complexity exists regarding Fitzpatrick, a relatively uncommon Irish surname, then it is likely there will be similar complexity among many Irish surnames, some of which have a single origin narrative.

## 6. The Untapped Fiants and Patents

For Irish seeking to discover their ancient origins, methods that focus on the attachment to a pedigree or a noble line fall short. Such methods are popular (Nash 2008), but a more critical approach is needed to make sense of the complexity of Y-DNA data from some Irish surname projects. The method used here is to align findings from the Fitzpatrick Y-DNA project with relevant archival data, those being two scarcely interrogated record sets: (a) the Fiants and (b) the Patents of Ireland, which are "an extraordinary and largely untapped source of information" (Fitzpatrick 2021, p. 66). The method has led to much positive discovery for some Fitzpatricks, but discomfort for others.

The 'Letters Patent' issued under the Great Seal of Ireland provides records such as grants, leases, and pardons. The Fiants, which are also records of the Letters Patent, are so named from the wording of the original legal documents—"Fiant literæ patentes", i.e., 'let the letters patent be' (Deputy Keeper of the Public Records in Ireland 1875, p. 11). The Fiants were once considered of lesser importance than the Patents, serving only to direct the Letters Patent to be passed. However, the Fiants are now considered superior since although they "were all supposed to be enrolled on the Patent Rolls" numerous instances occur where the Patent is not found (Morrin 1861, p. xxxiii); hence, the Fiants can provide more information than the Patents. Moreover, many Patents have been lost (Fewer 2019).

This article taps into the Fiants and Patents related to Fitzpatrick and the Gaelic surnames from which it was derived, most notably Mac Giolla Phádraig or Ó Maol Phádraig, which were among many Irish surnames subjected to "the mutilation and corruption . . . that took place in the seventeenth and to a lesser extent in the eighteenth centuries" (MacLysaght 1985, p. 23). It is notable that the corruption of both Gaelic forms resulted in the loss of the important descriptors Giolla (meaning servant, or follower) and Maol (meaning tonsured, or devotee). Hence, Mac Giolla Phádraig (son of the servant of [St] Patrick) or Ó Maol Phádraig (descendant of the devotee of [St] Patrick) both became Fitzpatrick (meaning, simply, son of Patrick), which is not a true anglicised version of either of the Gaelic forms, since it is stripped of the association with adherence to St. Patrick.

As well as surnames, the Fiants and Patents also record patronymic placenames, many of which are now similarly corrupted or entirely lost, yet they do much to inform Fitzpatrick, and other Irish surname, genealogy.

For this article, the Fiants reviewed were from 1521–1603, and Patents from 1514–1575 and 1603–1633, i.e., much of the reign of Henry VIII of England to the eighth year of Charles I of England; the Fiants and Patents from this period are voluminous and provide rich pickings; it was also a period when the British colonial project in Ireland had gained much momentum (Rahman et al. 2017). The method was straightforward: (1) identify all occurrences of 'Patrick' (or similar forms, such as Patricke and Patraic) used as a part of a surname in Fiants and Patents; (2) if possible, collate the surname appearances according to location, e.g., parish, or townland; (3) correlate the archival locations with the origins of the most distant known ancestor (MDKA) of members of the Fitzpatrick DNA project.

As will be demonstrated, it is not uncommon for placenames in the Fiants and Patents to be challenging to pin down to modern locations; fortunately, the 'Placenames Database of Ireland' (https://www.logainm.ie/en/, accessed on 4 July 2023) provides excellent summaries of archival records that can be of aid. In this article we provide three examples of Fitzpatrick clans that have been uncovered using this method.

### 7. The Fitzpatrick Dál gCais: A Lost Clan Found

The Fitzpatrick Dál gCais are named so here first on a genetic basis. They are of Y-haplotype R-Z253 . . . ZZ31_1, which is estimated to have arisen ca. 500 AD (Fitzpatrick et al. 2022). R-Z31_1 is the Y-haplotype of the late Sir Conor O'Brien, a direct descendant of Brian Bóruma, High King of Ireland (Wright 2009; Swift 2014). Bóruma was a descendant of Cas, a fifth-century semi-mythical Munster king whose ancestors became known as Dál gCais (or Dalcassians, 'of the tribe of Cas'). Dalcassian clans rose to great prominence in Thomond (effectively modern-day Co. Clare and Co. Limerick) and, by the tenth century, had seized the entire Kingdom of Munster and even pressed into the neighbouring Kingdom of Ossory, to the east (Byrne 1973).

With the progression of commercial Y-DNA testing into the era of next-generation sequencing (NGS), there came an understanding that groups of Fitzpatricks are Dál gCais. Earlier Y-DNA testing had already identified them as a distinct Fitzpatrick group. However, because they were not associated with the dominant Fitzpatrick group, two explanations for their origins were offered: they were either descended "from a survivor of the wreck of the Spanish Armada off the coast of Co. Clare in 1588, or else their ancestor was a Viking who left a genetic souvenir during a pit stop in the area" (The Fitzpatrick DNA Study 2007). Looking back, it is easy to criticise both explanations for being poorly considered. Yet, here is a poignant reminder that, in the early days of Y-DNA research, ill-based 'findings' depreciated the identity of others. At the time of the Spanish/Viking theory, members of the Fitzpatrick Dál gCais were not attached to a common narrative and traced their geographic origins to diverse locations (Co. Clare, Co. Galway, Co. Mayo, and Co. Roscommon); there was no account or memory of a common clan ancestry or how the clan came to be dispersed. The Fiants and Patents now prove a rich record source via which the narrative of the Fitzpatrick Dál gCais clan can be reconstructed and understood.

Fiants 4753, 4805, and 4806 of Elizabeth I, dated 1585–86, issue pardons to numerous individuals and among those named are "Teig M'Gillephadrick, of Legarde [Leagard], yeoman, Dermod m' Rorie M'Gillephadrick, of Mylagh (Moy) . . . Diermod m'Fynin M'Gyllepahdrick of Karwchell [Carrowkeel] . . . Diermod m'Donill M'Gillephadrick of Legarde" and "Scalan m'Teige M'Gillepatricke" (Deputy Keeper of the Public Records in Ireland 1883). These entries are a good example of the challenges in identifying placenames (modern names provided in brackets). Significantly, from a historical perspective, the records demonstrate that there were prominent Mac Giolla Phádraig in the Barony of Ibrickan, Co. Clare, in the late sixteenth century. Associates pardoned alongside the Mac Giolla Phádraig include Sir Turlagh O'Brien, the O'Flaherty of Aran Islands, and Mac Sweeny, Lynch, and Kirwan of Co. Galway. Sir Turlogh (1555–1623), a grandson of Connor O'Brien, King of Thomond (1528–1540), was a high-ranking family member (Cunningham 1994); the O'Flaherty had "the lordship and rents of Aran" (Lynam 1914); the Mac Sweeney were well-known gallowglass, i.e., elite mercenaries, (Walsh 1920); and the Lynch and Kirwan were two of several merchant families of Galway Town who were known as 'The Tribes' and were "classed amongst the most considerable merchants of Europe" (Hardiman 1846, p. 20). Hence, the Fiants provide a crucial record of a Fitzpatrick clan and their place in society from which it is possible to perform further gene-alogy.

The pardons issued were part of the 'Composition of Connaught', a colonising strategy that is described as the best example of the policy of surrender and regrant under Elizabeth I, it being "the great settlement of the landed property in Clare and Connaught" (Butler 1925, p. 106). McInerney (2011) explains that the Composition aimed to replace payments exacted from the common people of Connaught by Irish lords and the English alike, with

a land rental, as "part of a wider push of spreading English law and [encouraging] local sept-heads to pursue freeholder status and break the client-patron dependence with their overlords" (p. 3). In 1585 the Mac Giolla Phádraig of Co. Clare were clients of the Earls of Thomond and among their most "important retainers and followers" (McInerney 2013, p. 175). At the 'Composition of Connaught', the Earl of Thomond claimed the entirety of Ibrickan. Although they probably did not foresee it, the Mac Giolla Phádraig would soon be evicted from their lands. The surrender and regrant policy was a "concealed system of confiscation" (Butler 1913, p. 100), and the Uí Briain moved to exploit their "kin and clientage network" and increase their landholding (McInerney 2018, p. 62). Moreover, those clients later deemed rebels against the Crown found the "whole possessions of the clan could be seized" (Butler 1925, p. 218).

After 1585, the Mac Giolla Phádraig established networks beyond Co. Clare. An estate record from 1605 shows Valentine Blake, of the merchant 'Tribes' of Galway, acted as attorney for Donnough O'Brien, Fourth Earl of Thomond, concerning lands in the Barony of Corcomroe; Fynin Mac Giolla Phádraig of Doonsallagh was a witness (Ainsworth 1961). Additionally, between 1618 and 1621, Fynin is recorded with Oliver Martin of 'The Tribes', conveyed lands in the Barony of Burren (McInerney 2013). The Mac Giolla Phádraig had set themselves in the orbits of the Galway merchant 'Tribes', perhaps sensing a coming calamity, and so it transpired. At the well-documented 1642 siege of Tromra, Peter Ward, an English settler, was killed by men under the command of Colonel Edmund O'Flaherty, who was afforded "all manner of necessary instantly upon his landing" by the Seneschal of Ibrickan, Richard Fitzpatrick (Trinity College 1641), his surname now corrupted from Mac Giolla Phádraig. Initially, Fitzpatrick's employer, Sir Barnaby O'Brien, the Sixth Earl of Thomond, elevated him to his receiver and chief servant, yet pressure came to bear on the Earl, for whom the tension of choosing to side with protestant colonisers or catholic native Irish was very real, his allegiances "continually waver[ed] to suit the occasion" (Breen 2014, p. 15). Within one decade, the rebel Fitzpatricks had lost their lands. The last prominent member of the once proud Ibrickan clan was Fynin Fitzpatrick, a great-grandson of Fynin of Doonsallagh, who assigned Lisdoonvarna to Daniel O'Brien in 1652.

The Ibrickan Fitzpatricks relocated and prospered. By 1685 John Fitzpatrick and his son Richard held the lease of Aran at £500 per annum. There were notable intermarriages of these Fitzpatricks with members of 'The Tribes'. In 1686, John's son Richard married Joan French of Spiddle, Co. Galway (the daughter of Anthony French, Sheriff of Galway in 1654), and his daughter, Bridget, married George Morris, also of Spiddle, in 1696. Another son, Edmund, married Annabelle Martin; their son Richard was Sheriff of Galway in 1730, Mayor in 1738, Deputy Mayor in 1747, and a member of the Irish parliament who represented Galway from 1761 to 1767 (Hardiman 1820, 1846). The family remained prominent in Galway for a further 50 years; in 1797, Edward Fitzpatrick was Sheriff, and he was Deputy Mayor at the time of his death in 1817 (Fitzpatrick 1817), after which the Galway historian James Hardiman could not ascertain whether "any of the name now exist" (Hardiman 1846, p. 430).

However, there were other Fitzpatricks of Aran, seemingly unknown to Hardiman. On 3 November 1696, the will of Patrick Fitzpatrick, yeoman, of Aran, and father of the aforementioned John, was proved. On the same day, the will of Daniel Fitzpatrick of Aran, yeoman, was also proved. The latter gave administrative rights to his son, Moriertagh Fitzpatrick, who married Elenor Browne of 'The Tribes' in 1700 (Crosslé 2023), which evidences the Fitzpatricks were probably on Aran from at least the mid-seventeenth century, but if there were any descendants of Moriertagh on Aran or in Galway town, then any record of them is lost. It is suggested that some Fitzpatricks of Aran and Galway may have relocated and that further colonising forces played a part. After Galway was taken by Cromwell's forces in 1652, there came an eviction from the town of many families of 'The Tribes'. They were denied the restoration of their estates by the 1665 'Act of Explanation', but reinstations and compensations were made. Notable concerning this article were lands outside of Galway granted to the families of Blake and French.

In 1677, under the Act of Settlement, Francis Blake of Merlin Park, Galway, received a grant for Ballyglass, Co.Mayo (Blake 1905), where Fitzpatrick families were domiciled by at least the late eighteenth century. Furthermore, since Fitzpatrick (Mac Giolla Phádraig) is not an Irish clan name associated with Mayo, the arrival of Fitzpatricks in Ballyglass may have been part of a Blake resettlement. That Ballyglass Fitzpatricks are Y-haplotype R-Z31_1 joins them to their genetic cousins found today in Co. Galway and Co. Clare, and back to the Dál gCais, their narratives lost through colonising moments, are now re/membered.

An extant line of Fitzpatricks from Tiboine parish, Co. Roscommon, also R-Z31_1, can be similarly accounted for. Under the Act of Settlement, Dominick French secured a grant of considerable portion of land in the Barony of Boyle (D'Alton 1847), which was Mac Dermott's Country (Freeman 1936) and never the territory of any Mac Giolla Phádraig clan. The southwest portion of Boyle was cleaved off to form the Barony of Frenchpark, which included lands in Tibohine, where Fitzpatricks are found from the mid-eighteenth century. The 1749 'Census of Elphin', which was a religious survey of the diocese organised by Bishop Edward Synge (Legg 2004), lists four Fitzpatrick households in Tibohine, and all four recorded male heads of household were Smiths, an occupation that continued with Fitzpatricks there until at least the time of Griffith's Valuation when Michael Fitzpatrick is recorded as leasing a house and forge (Griffith 1864).

It is worth noting that the spectre of colonisation has also clouded the true origins of the Fitzpatrick Dál gCais in ways other than the loss of memories. Colonisation brought with it a clamoring among some Fitzpatricks to be known as the descendants of the Fitzpatrick Barons of Upper Ossory, who first gained that title from Henry VIII via surrender and regrant (Maginn 2007), and even to be considered the rightful heir to the defunct title; this led to a least one spurious pedigree claim, which served to confuse the true origins of the Fitzpatrick Dál gCais. The 'Statement of the Pedigree of Nicholas Fitzpatrick, Esquire, Claiming the Title, Honour, and Dignity, of Baron of Upper Ossory' (National Library of Ireland n.d.), is such an example—Nicholas traced his descent from the same John Fitzpatrick of Aran and claimed John was the son of Denis, a descendant of Bryan, first Baron of Upper Ossory. The claim was easily dismissed since John was the son of Patrick, according to the latter's will. Notably, the legal counsel who prepared the 'Statement of Pedigree' was Henry Nugent Bell, known for his "suspicious luxuriance of imagination" (Goodwin 2004).

Inaccurate associations are another way in which Mac Giolla Phádraig clans have been subject to the impacts of colonisation. Remarkably, a link between Fitzpatrick Dál gCais and Upper Ossory has substantial merit. Fynin Fitzpatrick, of Lisdoonvarna, understood his Mac Giolla Phádraig pedigree back more than four hundred years. The funeral entry of his father, Dermot (d. 1637), traces him back to Connor, "second brother of Daniell more laterlie called mcGullepatrick Lord of Upper Ossory" (National Library of Ireland 1638). The great Roger O'Farrell went further; he recorded that Connor, who "settled in Thomond", was the son of Scanlan, a late twelfth-century Mac Giolla Phádraig chieftain of Upper Ossory, and provided an unbroken lineage to Dermot, Scanlan's twelfth great-grandson (O'Farrell 1709). Hence, Dál gCais Fitzpatricks, who are Y-haplotype R-Z31_1, have a sound narrative; today, they may be located in Clare–Aran–Galway–Mayo–Roscommon, but they were once of Ossory. Their Y-DNA, with help from the Fiants, brings their origins back. However, as will be discussed, this disrupts the narrative of other Fitzpatrick clans who consider themselves to be the descendants of Upper Ossory Mac Giolla Phádraig.

## 8. The Lost Clan Places

Important in the decolonisation of Irish clans is not only rediscovering those who had seemingly ceased to exist, but also reconnecting those who have lost the knowledge of their ancient homelands. For many of the Irish diaspora, a pilgrimage home to their ancestors' townlands is an integral part of their life's journey. However, what if the townland, parish, or even county is unknown, and all that is known is that they are Irish? The finding of place has been a recurrent theme among Fitzpatricks on the DNA project. Y-DNA

analysis, in conjunction with Fiants and Patents, has uncovered narratives that will offer encouragement to other Irish clans on similar journeys of locating the place/s their clan was connected to. Here, we provide three examples, which also demonstrate contrasting outcomes relating to personal and clan identity.

### 8.1. Mac Giolla Phádraig's Meadow

The narrative we have presented of the Fitzpatrick Dál gCais touched on those who lived in Ibrickan, in west Co. Clare, who can now stand at the Cliffs of Moher, on the lands of their Mac Giolla Phádraig ancestors. However, Mac Giolla Phádraig lived in other parts of Co. Clare, although they are not individually named in Fiants or Patents. A 1621 Patent details the grant of Co. Clare lands from James I to Donnough O'Brien, the Fourth Earl of Thomond, which included the half-quarter of "Ballyclenymcgillapatrick" (Griffith 1966, p. 493). The name of the small townland (a sub-denomination) possibly stems from Baile Chluaine Mac Giolla Phádraig, i.e., the settlement of "Mac Giolla Phádraig's watery meadow" (L. McInerney, pers.comm., 18 January 2022). A "lesser possibility is that it is that [cleny] is a mangled form of clann" rendering it Baile Chlainne Mac Giolla Phádraig, i.e., the settlement of Clan Mac Giolla Phádraig (L. McInerney, pers.comm., 15 July 2023). Although the exact location of 'Ballyclenymcgillapatrick' is not yet known, the general area is understood from other townlands listed in the Patent; it was probably in the parish of Inchicronan in northeast Co. Clare.

### 8.2. Mac Giolla Phádraig's Farm

The Fitzpatrick of Leinster have been defined as such only relatively recently. Once considered to have only a limited regional presence in Co. Down, the clan is now believed to have an ancient Leinster origin and a narrative of a sixteenth-century dispersion. It was not that the early classification of the clan, via Y-DNA and traditional genealogy, was flawed, but simply that, by ca. 2014, of the eight out of ten group members who knew their early-nineteenth-century origins, all traced to Co. Down; that locale, understandably, became their identifier. However, there was a hint of something more complex than a southeast Ulster origins narrative—an underlying Y-DNA signature referred to as the 'Leinster' or 'Irish Sea' modal haplotype that was considered, from the time of the 2006 Trinity DNA study, to be strongly associated with clan O'Byrne of Wicklow (McEvoy and Bradley 2006). With more than 30 participants on the Fitzpatrick Y-DNA project, the Fitzpatricks of Leinster are identified via NGS by the highly surname-specific Y-haplotype R-BY2849, which arose ca. 1100 AD. The early nineteenth-century origins of participants are still mostly Co. Down, notably in either the Baronies of Upper Iveagh or Mourne or the Lordship of Newry but are now also located in Co. Kildare, southern Co. Louth and Co. Carlow. However, neither clan nor the surname Fitzpatrick was known in Co. Down before the '1659 Census', when they were recorded as "McIlepatricke", of Upper Iveagh (Pender 1939, p. 77). So from where did these Mac Giolla Phádraig come?

Without the Fiants, there would be little hope of understanding the pre-seventeenth-century origins of the Co. Down branches of the Fitzpatrick of Leinster. However, R-BY2849 Fitzpatricks of Co. Kildare can now be traced via vital (i.e., birth, marriage, death) and land records to Monasterevin and the surrounding townlands. Indeed, there are living Fitzpatricks associated with a nearby townland today called Kilpatrick; yet, it was once named otherwise. A 1567 Fiant of Elizabeth I refers to the surrender by Edward Waterhouse, Chief Secretary for Ireland, of his manor and "lands of Evon" (i.e., Monasterevin), which included "Grange McGilpatrick", i.e., Mac Giolla Phádraig's farm, which once belonged to the Monastery of St. Evon, and later to the Manor House of Monasterevin. Two years later, the townland's name was recorded as "Kilpatricke" (Deputy Keeper of the Public Records in Ireland 1879). Further, a 1620 Patent refers to Kilpatrick as "alias Grange McGilpatrick" (Griffith 1966, p. 474), which is linked to the presence of an ancient Mac Giolla Phádraig clan that once resided near Monasterevin.

The Fiants also provide a possible reason why some Fitzpatricks of Leinster may have moved to Ulster. In 1611, near the top of a list of pardons granted to Ever Mac Shane Mac Owen O'Neill of Edenduff-Carrick, County Antrim, and his kith and kin, is Neece McGilpatrick (Griffith 1966), who stands out as the only person of that surname. This Neece (i.e., Naos, a pet form of Aónghus) may be the same person referred to in the Bagnall Estate records of 1688 for Ballygowan "formerly sett to Neece McGilpatrick the smith and now left to his son Owen and the rest" (Public Record Office of Northern Ireland 1688). It is plausible, therefore, that some Leinster Mac Giolla Phádraig assisted the O'Neill during the Nine Year's War and chose to remain in Ulster afterwards.

Fast forward from 1688, and various vital and land records from the late eighteenth and early nineteenth century find numerous Fitzpatrick families in Co. Down. Notable is an 1803 agricultural census, which stemmed from concerns over food supply and "fears of French invasion" (Turner 1984, p. 19). Parallel records exist, some entirely handwritten and others filled in on a pre-printed form. Notable is a record in the parish of Maghera. In the fully handwritten summary, the recorder notes down Daniel and Lawrence Mcllepatrick. However, a parallel record on a pre-printed form has the same Daniel and Lawrence as 'Fitzpatrick'. Hence, while no living memory of or archival record of Leinster origins has yet been found for Co. Down Fitzpatricks, it is clear the memory of the ancient Leinster surname Mac Giolla Phádraig, corrupted to Fitzpatrick, lingered in the minds of some.

*8.3. The Town of the Mulpatrick*

Another group of Fitzpatricks who trace broadly to Co. Cork who belong to Y-haplotype R-CTS4466 is considered to be an Eoghanachta (i.e., the descendants Éogan, a third century Munster king; Byrne 1973) since it correlates strongly with the surname O'Sullivan (McEvoy and Bradley 2006). Woulfe (1923) noted the patronymic surname O'Mulpatrick (Ó Maol Phádraig) was once a common surname, especially in Cavan and Cork, yet it is now considered extinct. The Annals of Ulster record Máel Pátraic, King of Uí Liatháin, a territory nearly co-extensive with the present Barony of Barrymore, Co. Cork, in 944 AD (Hennessy 1866, 1887); hence, it is undoubtedly possible an O'Mulpatrick clan may have arisen in ancient times from him. Moreover, while the Fiants and Patents offer no record of Mac Giolla Phádraig of Co. Cork that could explain why the surname Fitzpatrick occurs, there are records of several O'Mulpatrick. Two sons of Connor O'Mulpatrick are recorded between 1577 and 1585, along with a placename—Rahanisky in the Barony of Cork (Deputy Keeper of the Public Records in Ireland 1883). Furthermore, three Fiants from 1601 record eight individual O'Mulpatricks in the Baronies of Fermoy, Carbery East, Kilmore, and Imokilly (Deputy Keeper of the Public Records in Ireland 1885; Griffith 1966), demonstrating they were widespread in Co. Cork in the sixteenth and seventeenth centuries. Notably, the Co. Cork O'Mulpatricks recorded in the Fiants are alongside those of Eoghanachta surnames, such as the O'Sullivan and Mac Carthy, hinting at kinship and an older common shared ancestry.

Highly significant regarding the Co. Cork occurrences of O'Mulpatrick is a 1568 Patent of Elizabeth I and a 1609 Fiant of James I, which record the placename "Ballymulpatrick", i.e., the town of the Mulpatrick, in the Barony of Imokilly (Griffith 1966, p. 130); this indicates the O'Mulpatricks of Co. Cork were likely no fleeting visitors. There is no record of Ballymulpatrick in the Down Survey of Ireland (ca. 1656), but from the survey maps, it can be deduced that the lands where it once was had become called Ballymacpatrick (http://downsurvey.tchpc.tcd.ie, accessed on 7 July 2023), which by the time of the Ordinance Survey of Ireland (ca. 1838) had been subsumed by the adjacent townland of Ballymacandrick—no record of Ballymulpatrick or Ballymacpatrick is found in definitive placename books, such as the index associated with Petty's barony maps (Goblet 1932), and the 'Placenames Database of Ireland' has no account of it. Hence, without the Fiants and Patents, the O'Mulpatrick of Co. Cork and the place where some of them once lived would be lost. Yet, among the discovery are cautionary notes.

### 9. What Do We Learn?

When working with Y-DNA in conjunction with Fiants and Patents, it is possible to discover lost Fitzpatrick clans. So, we ask the question, how did they become lost? We argue the colonising project and associated strategies, including a dominant narrative that said all Fitzpatricks were the same, served to erase their memory over time. However, Y-DNA brought a fresh impetus to look and provided a provocation to examine archival sources more deeply and critically, to work the gene-ealogical hyphen. As with much gene-ealogical work, historically written histories cannot be solely relied on. This work demonstrates the value of the much-underutilised Fiants and Patents as records that can enrich surname and clan narratives.

The reimagined narrative of the Fitzpatrick Dál gCais highlights the way colonisation, in its many shapes and forms, worked to erase these connections over time. Once prominent in Co. Clare, several political events meant the clan became lost. The loss of lands, loss of their surname, corruption to Fitzpatrick, loss of connection to an ancient clan identity, loss of a language, and the dispersal to other counties and countries all contributed, by the mid-nineteenth century, to doubts of whether the clan even still existed. Exist they do, and there is a delight among members of the Fitzpatrick Dál gCais in being re/membered, as evidenced through becoming registered as a clan with the Clans of Ireland in 2022.

Yet, while there has been delight for some Fitzpatricks, for others the complexities of Y-DNA studies in relation to identity have been highlighted. During the early years of the Fitzpatrick Y-DNA project, one genetic group was considered the descendants of the barons of Upper Ossory and referred to as 'The Noble Line' based on their origins being in Co. Laois and Co. Kilkenny. Disruption came when some who claimed to descend from the barons were found to be Y-DNA haplotype J-FTA78391, whose members held a robust pedigree, while others with the same claim, whose pedigree claim was weaker, were found to be haplotype R-A1488. Both genetic groups are represented in the clan Fitzpatrick of Upper Ossory, registered with Clans of Ireland, who, in their wisdom, state "the science of DNA is still evolving and membership of an Irish Clan is based on one's inherited and chosen identity and not on bloodline descent alone" (Clans of Ireland 2023).

However, the complexity for clan Fitzpatrick of Upper Ossory is more than just two genetic groups claiming the same paternal lineage. The barons were said to descend from the Mac Giolla Phádraig of Ossory, an ancient Irish clan. J-FTA78391 is associated with Levantine populations (El-Sibai et al. 2009), whereas R-A1488 is associated with Norman families and, earlier, with Scandinavia (Rodríguez-Varela et al. 2023). Hence, as well as dealing with the complexity of competing paternal lines, there is also the disruption of a connection to ancient Irish origins. Here, then, is an excellent example of a disrupted identity that was previously narrated with links to an ancient Irish clan dynasty, a narrative that has served the clan well over several generations. The practice of pedigree genealogies was essential to British colonial structures of control. The practice of amateur genealogists searching for ancestors of note has further inflated the importance of colonial pedigree genealogies.

The dominant narrative of the barons of Upper Ossory has been reinforced by those wishing to claim a connection to an ancient pedigree, who hope to find an Irishness outside of the white settler identity they encounter in the place they occupy. Gene-ealogy, by employing Y-DNA data in conversation with archival data, establishes their origins outside of ancient Ireland. In contrast, another clan, the Fitzpatrick Dál gCais, has a legitimate origin with the Mac Giolla Phádraig of Ossory. How does a clan such as the Fitzpatricks of Upper Ossory work through such disruptions to an ancient narrative that has served them well? Not easily, and this highlights the dangers of investing heavily in an Irish clan definition defined by paternal lineage, which is also reliant on traditional and dominant narratives. The disruption of Irish clan identities warrants further investigation, but recognising that the clan Fitzpatrick of Upper Ossory has been Irish for hundreds of years is important. They are Irish; Y-DNA, of course, can only give us one story of many.

The 'good vibes' gene-ealogy has brought to the R-BY2849 Fitzpatricks of Leinster are understandable—kindred connections have been made across distant lands and across

the ages. Ancient homelands have been rediscovered, and the clan's name, Mac Giolla Phádraig, which perhaps was never truly lost, has been reaffirmed. This is the stuff many genealogists dream of. Why, then, has there yet to be a cry of excitement from the R-CTS4466 Fitzpatricks, who are quite possibly of a lost clan O'Mulptrick of Co. Cork? Identity, how it is ascribed, and how it is chosen are tricky things. When there are histories that have been erased over time, deliberately or consequential of ongoing colonising strategies, how people define themselves needs to be respected. Economic, social, religious and political reasons can come into play. What might benefit one individual from claiming a historical clan name might disrupt another's chances with the life they are comfortable with or yearn for. For the R-CTS4466 Fitzpatricks of Co. Cork, we can only assume, but imagine if your family Fitzpatrick had built over several generations a thriving, respectful business in the county under the name Fitzpatrick, and you are celebrated for the cultural identity you bring, identifiable through your surname. How would it feel if some enthusiastic scholar gene-ealogist knocked on the door one day with the 'great news' that you may be a descendant of Máel Pátraic, a tenth-century king, or some sixteenth-century O'Mulpatrick rebels? Perhaps a 'Fitzpatrick family business' has barely survived juggling economic tensions through Covid lockdowns or has been dealing with increasing complexities to Irish identity with the influx of immigrants to a small community. Fitzpatrick is a name that gives security, strength, and stability. It is a name that the Fitzpatricks of Co. Cork have grown up with over many generations; it is the history that is known, and which has agency. O'Mulpatrick is utterly unknown.

Belonging and being proud of the kinship group one associates with is vital to self-efficacy and well-being. Increasingly, settler groups are called on to take account of themselves and their histories concerning indigenous inequities, racism, and histories of practices of erasure. Ireland is no different. With a turbulent history of encountering ongoing practices of colonisation and a population today of peoples who are mixed native Irish/colonial settlers, and increasing immigration with new settlers or displaced peoples, how one belongs as Irish is complicated. Also complicated are those diaspora people who yearn to claim 'Irishness' in their places as something distinct from colonial settlers.

Hence, working the hyphen between Y-DNA data and Fiants and Patents has been rewarding for some Fitzpatricks but challenging for others. It has unfolded connections to clan histories and kinships that were previously lost via colonisation, but has also disrupted dominant narratives and clan identities. Even a cursory look at the Fiants and Patents reveals the occurrence of many place names of the form 'Bally-surname' that now appear to be 'lost'. Hence, it is possible that decolonisation narratives could be uncovered for other Irish clans, along with the potential for the disruption of clan identities.

**Author Contributions:** Both authors contributed equally to all elements of the research and writing of this article. All authors have read and agreed to the published version of the manuscript.

**Funding:** This research received no external funding.

**Institutional Review Board Statement:** This study did not require ethical approval, as it involved the review of publicly available data.

**Informed Consent Statement:** Not applicable.

**Data Availability Statement:** Publicly accessible data from the Fitzpatrick Surname Project can be viewed at the Family Tree DNA hosted website https://www.familytreedna.com/public/fitzpatrick?iframe=yresults, accessed on 7 August 2023.

**Acknowledgments:** The authors thank genetic genealogist Ian Fitzpatrick for his many hours working to help understand Fitzpatrick Y-DNA and clan connections. Thanks also to Karen Fitzpatrick Hall of the Dál gCais clan; Sharon Fitzpatrick, Marianne Mielke, and Maureen Arthur of the Upper Ossory clan. This article stemmed from a presentation given by one of us (Mike Fitzpatrick) at the Clans of Ireland Cultural Summit, May 2023 (https://www.youtube.com/watch?v=UtdDw4Kk0vk, accessed on 17 July 2023); the support of Clans of Ireland, particularly Gearóid O Ceallaigh and Luke McInerney, is acknowledged.

**Conflicts of Interest:** The authors declare no conflict of interest.

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
