# Peer review of "Decolonising an Irish Surname by Working the Hyphen of Gene-Ealogy"

_genealogy, doi:10.3390/genealogy7030058_

Round 1
Reviewer 1 Report
This study compares Y-DNA lineage data with rarely consulted archival data to bring new light to the origins and mutations of variants of the Fitzpatrick surname, using this as an example of how "gene-ealogy" can be worked as a means of decolonising settler colonial histories. The study's material is scrupulously researched and carefully mapped out, and will undoubtedly be valuable for other historians and genealogists interested in the history of Fitzpatrick surnames, or in applying a similar methodology to other names and contexts. My only suggestions (though these are not obligatory) would be the following:
1) to include a comment (e.g. p.5 of manuscript) about why "Genghis Khan" type myths (which ascribe a given DNA lineage or surname to one, particularly prolific, male ancestor) tend to hold so much sway in the collective imagination and to be uncritically accepted – see C. Nash Genetic Geographies (2015) for a feminist analysis of this tendency.
2) add a note explaining what the authors understand by the phrase "genetically Celtic" (p.6), since various archaeologists and other scholars have consistently objected to the idea that ethnicity can be mapped onto or ascribed by genetic lineages. See for example C. Nash "Irish Origins, Celtic Origins: Population Genetics, Cultural Politics" (2006); C. Frieman & D. Hofmann "Present pasts in the archaeology of genetics, identity, and migration in Europe: a critical essay" (2019).
Otherwise, this is an excellent article and I recommend it for publication without further modifications.
Author Response
Thank you for the comments regarding our manuscript we are excited to see it being processed to this stage. We appreciate the encouraging words from both reviewers and the thoughtful suggestions for enhancing the work. Below are the changes we have made which are highlighted as advised on the revised manuscript.
Line 12: fixed typo;
Lines 28-29: corrected reference;
Line 96: corrected reference;
Line 154: word added;
Line 168: large quote edited;
Line 169: clarified sept meaning;
Line 172: added quotation marks;
Line 176: fixed typo;
Line 197-199: added clarity around ‘Genghis Khan’ concept.
Lines 241-244: added clarity around defining Celts on a genetic basis;
Lines 290-295: added clarity on the anglicisation to Fitzpatrick;
Line 331: fixed typo;
Line 351: defined gallowglass;
Line 382-389: expanded (not based on peer-review comments)
Line 548: removed self-citation;
Lines 608-609: removed self-citation;
Lines 725-731: two Fitzpatrick citations removed.
Line 780: Nash reference added;
Lines 800-801: Scully reference added.
Ngā mihi
Esther and Mike Fitzpatrick
Reviewer 2 Report
This is a well theorized piece of genealogical work, covering interpretational problems and avenues leading towards their resolution.
It would benefit from a brief early discussion about the linguistic tension between the English and Irish forms of the surname in question, which do not match (what has happened to giolla during anglicization?)
Some terms may not be known to general readers (gallowglass) and others perhaps could be reviewed for a specific definition in the context of this paper (clan, sept).
There are a very few typos that can be found by search (deolonisation, disupt, poingant).
See above.
Author Response

(The authors gave the same response as above.)
